# Conservative Data Sharing
# for Multi-Task Offline Reinforcement Learning

**Tianhe Yu**[*,1,2], **Aviral Kumar**[*,2,3], **Yevgen Chebotar**[2], **Karol Hausman**[1,2],
**Sergey Levine**[2,3], **Chelsea Finn**[1,2]

[1]Stanford University, [2]Google Research, [3]UC Berkeley       (*Equal Contribution)
tianheyu@cs.stanford.edu, aviralk@berkeley.edu

## Abstract

Offline reinforcement learning (RL) algorithms have shown promising results in domains where abundant pre-collected data is available. However, prior methods focus on solving individual problems from scratch with an offline dataset without considering how an offline RL agent can acquire multiple skills. We argue that a natural use case of offline RL is in settings where we can pool large amounts of data collected in various scenarios for solving different tasks, and utilize all of this data to learn behaviors for all the tasks more effectively rather than training each one in isolation. However, sharing data across all tasks in multi-task offline RL performs surprisingly poorly in practice. Thorough empirical analysis, we find that sharing data can actually exacerbate the distributional shift between the learned policy and the dataset, which in turn can lead to divergence of the learned policy and poor performance. To address this challenge, we develop a simple technique for data-sharing in multi-task offline RL that routes data based on the improvement over the task-specific data. We call this approach conservative data sharing (CDS), and it can be applied with multiple single-task offline RL methods. On a range of challenging multi-task locomotion, navigation, and vision-based robotic manipulation problems, CDS achieves the best or comparable performance compared to prior offline multi-task RL methods and previous data sharing approaches.

## 1   Introduction

Recent advances in offline reinforcement learning (RL) make it possible to train policies for real-world scenarios, such as robotics [32, 60, 33] and healthcare [24, 67, 35], entirely from previously collected data. Many realistic settings where we might want to apply offline RL are inherently *multi-task* problems, where we want to solve multiple tasks using all of the data available. For example, if our goal is to enable robots to acquire a range of different behaviors, it is more practical to collect a modest amount of data for each desired behavior, resulting in a large but heterogeneous dataset, rather than requiring a large dataset for every individual skill. Indeed, many existing datasets in robotics [17, 11, 66] and offline RL [19] include data collected in precisely this way. Unfortunately, leveraging such heterogeneous datasets leaves us with two unenviable choices. We could train each task only on data collected for that task, but such small datasets may be inadequate for good performance. Alternatively, we could combine all of the data together and use data relabeled from other tasks to improve offline training, but this naïve data sharing approach can actually often degrade performance over simple single-task training in practice [33]. In this paper, we aim to understand how data sharing affects RL performance in the offline setting and develop a reliable and effective method for selectively sharing data across tasks.

A number of prior works have studied multi-task RL in the *online* setting, confirming that multi-tasking can often lead to performance that is worse than training tasks individually [56, 62, 90].

35th Conference on Neural Information Processing Systems (NeurIPS 2021).

These prior works focus on mitigating optimization challenges that are aggravated by the online data generation process [64, 89, 88]. As we will find in Section 4, multi-task RL remains a challenging problem in the offline setting when sharing data across tasks, even when exploration is not an issue. While prior works have developed heuristic methods for reweighting and relabeling data [3, 16, 44, 33], they do not yet provide a principled explanation for why data sharing can hurt performance in the offline setting, nor do they provide a robust and general approach for selective data sharing that alleviates these issues while preserving the efficiency benefits of sharing experience across tasks.

In this paper, we hypothesize that data sharing can be harmful or brittle in the offline setting because it can exacerbate the distribution shift between the policy represented in the data and the policy being learned. We analyze the effect of data sharing in the offline multi-task RL setting, and present evidence to support this hypothesis. Based on this analysis, we then propose an approach for selective data sharing that aims to minimize distributional shift, by sharing only data that is particularly relevant to each task. Instantiating a method based on this principle requires some care, since we do not know a priori which data is most relevant for a given task before we've learned a good policy for that task. To provide a practical instantiation, we propose the conservative data sharing (CDS) algorithm. CDS reduces distributional shift by sharing data based on a learned conservative estimate of the Q-values that penalizes Q-values on out-of-distribution actions. Specifically, CDS relabels transitions when the conservative Q-value of the added transitions exceeds the expected conservative Q-values on the target task data. We visualize how CDS works in Figure 1.

The main contributions of this work are an analysis of data sharing in offline multi-task RL and a new algorithm, *conservative data sharing* (CDS), for multi-task offline RL problems. CDS relabels a transition into a given task only when it is expected to improve performance based on a conservative estimate of the Q-function. After data sharing, similarly to prior offline RL methods, CDS applies a standard conservative offline RL algorithm, such as CQL [39], that learns a conservative value function or BRAC [82], a policy-constraint offline RL algorithm. Further, we theoretically analyze CDS and characterize scenarios under which it provides safe policy improvement guarantees. Finally, we conduct

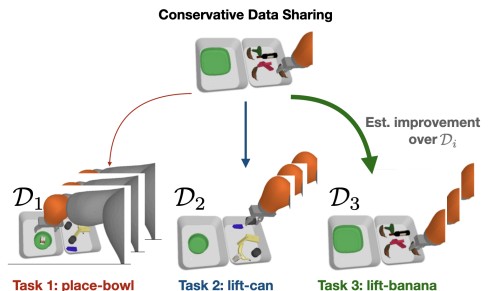

Figure 1: A visualization of CDS, which routes a transition to the offline dataset $\mathcal{D}_i$ for each task $i$ with a weight based on the estimated improvement over the behavior policy $\pi_\beta(\mathbf{a}|\mathbf{s}, i)$ of $\mathcal{D}_i$ after sharing the transition.

extensive empirical analysis of CDS on multi-task locomotion, multi-task robotic manipulation with sparse rewards, multi-task navigation, and multi-task imaged-based robotic manipulation. We compare CDS to vanilla offline multi-task RL without sharing data, to naïvely sharing data for all tasks, and to existing data relabeling schemes for multi-task RL. CDS is the only method to attain good performance across all of these benchmarks, often significantly outperforming the best *domain-specific* method, improving over the next best method on each domain by **17.5%** on average.

## 2   Related Work

**Offline RL.** Offline RL [14, 61, 40, 43] has shown promise in domains such as robotic manipulation [32, 52, 60, 70, 33], NLP [29, 30], recommender systems & advertising [72, 22, 7, 78, 79], and healthcare [67, 80]. The major challenge in offline RL is distribution shift [20, 37, 39], where the learned policy might generate out-of-distribution actions, resulting in erroneous value backups. Prior offline RL methods address this issue by regularizing the learned policy to be "close" to the behavior policy [20, 50, 29, 82, 93, 37, 68, 57], through variants of importance sampling [59, 74, 49, 75, 54], via uncertainty quantification on Q-values [2, 37, 82, 43], by learning conservative Q-functions [39, 36], and with model-based training with a penalty on out-of-distribution states [34, 91, 53, 4, 76, 60, 42, 92]. While current benchmarks in offline RL [19, 25] contain datasets that involve multi-task structure, existing offline RL methods do not leverage the shared structure of multiple tasks and instead train each individual task from scratch. In this paper, we exploit the shared structure in the offline multi-task setting and train a general policy that can acquire multiple skills.

**Multi-task RL algorithms.** Multi-task RL algorithms [81, 56, 77, 15, 27, 89, 85, 88, 33, 71] focus on solving multiple tasks jointly in an efficient way. While multi-task RL methods seem to provide a

promising way to build general-purpose agents [33], prior works have observed major challenges in multi-task RL, in particular, the optimization challenge [27, 64, 89]. Beyond the optimization challenge, how to perform effective representation learning via weight sharing is another major challenge in multi-task RL. Prior works have considered distilling per-task policies into a single policy that solves all tasks [62, 77, 23, 85], separate shared and task-specific modules with theoretical guarantees [13], and incorporating additional supervision [71]. Finally, sharing data across tasks emerges as a challenge in multi-task RL, especially in the off-policy setting, as naïvely sharing data across all tasks turns out to hurt performance in certain scenarios [33]. Unlike most of these prior works, we focus on the offline setting where the challenges in data sharing are most relevant. Methods that study optimization and representation learning issues are complementary and can be readily combined with our approach.

**Data sharing in multi-task RL.** Prior works [3, 31, 58, 63, 16, 44, 33, 8] have found it effective to reuse data across tasks by recomputing the rewards of data collected for one task and using such relabeled data for other tasks, which effectively augments the amount of data available for learning each task and boosts performance. These methods perform relabeling either uniformly [33] or based on metrics such as estimated Q-values [16, 44], domain knowledge [33], the distance to states or images in goal-conditioned settings [3, 58, 55, 48, 73, 47, 28, 51, 87, 8], and metric learning for robust inference in the offline meta-RL setting [45]. All of these methods either require online data collection and do not consider data sharing in a fully offline setting, or only consider offline goal-conditioned or meta-RL problems [8, 45]. While these prior works empirically find that data sharing helps, we believe that our analysis in Section 4 provides the first analytical understanding of why and when data sharing can help in multi-task offline RL and why it hurts in some cases. Specifically, our analysis reveals the effect of distributional shift introduced during data sharing, which is not taken into account by these prior works. Our proposed approach, CDS, tackles the challenge of distributional shift in data sharing by intelligently sharing data across tasks and improves multi-task performance by effectively trading off between the benefits of data sharing and the harms of excessive distributional shift.

## 3 Preliminaries and Problem Statement

**Multi-task offline RL.** The goal in multi-task RL is to find a policy that maximizes expected return in a multi-task Markov decision process (MDP), defined as $\mathcal{M} = (\mathcal{S}, \mathcal{A}, P, \gamma, \{R_i, i\}_{i=1}^N)$, with state space $\mathcal{S}$, action space $\mathcal{A}$, dynamics $P(\mathbf{s}'|\mathbf{s}, \mathbf{a})$, a discount factor $\gamma \in [0, 1)$, and a finite set of task indices $1, \cdots, N$ with corresponding reward functions $R_1, \cdots, R_N$. Each task $i$ presents a different reward function $R_i$, but we assume that the dynamics $P$ are shared across tasks. While this setting is not fully general, there are a wide variety of practical problem settings for which only the reward changes including various goal navigation tasks [19], distinct object manipulation objectives [83], and different user preferences [10]. In this work, we focus on learning a policy $\pi(\mathbf{a}|\mathbf{s}, i)$, which in practice could be modelled as independent policies $\{\pi_1(\mathbf{a}|\mathbf{s}), \cdots, \pi_N(\mathbf{a}|\mathbf{s})\}$ that do not share any parameters, or as a single task-conditioned policy, $\pi(\mathbf{a}|\mathbf{s}, i)$ with parameter sharing. Our goal in this paper is to analyze and devise methods for data sharing and the choice of parameter sharing is orthogonal, and can be made independently. We formulate the policy optimization problem as finding a policy that maximizes expected return over all the tasks: $\pi^*(\mathbf{a}|\mathbf{s}, \cdot) := \arg\max_\pi \mathbb{E}_{i \sim [N]} \mathbb{E}_{\pi(\cdot|\cdot, i)}[\sum_t \gamma^t R_i(\mathbf{s}_t, \mathbf{a}_t)]$. The Q-function, $Q^\pi(\mathbf{s}, \mathbf{a}, i)$, of a policy $\pi(\cdot|\cdot, i)$ is the long-term discounted reward obtained in task $i$ by executing action $\mathbf{a}$ at state $\mathbf{s}$ and following policy $\pi$ thereafter.

Standard offline RL is concerned with learning policies $\pi(\mathbf{a}|\mathbf{s})$ using only a given static dataset of transitions $\mathcal{D} = \{(\mathbf{s}_j, \mathbf{a}_j, \mathbf{s}'_j, r_j)\}_{j=1}^N$, collected by a behavior policy $\pi_\beta(\mathbf{a}|\mathbf{s})$, without any additional environment interaction. In the multi-task offline RL setting, the dataset $\mathcal{D}$ is partitioned into per-task subsets, $\mathcal{D} = \cup_{i=1}^N \mathcal{D}_i$, where $\mathcal{D}_i$ consists of experience from task $i$. While algorithms can choose to train the policy for task $i$ (i.e., $\pi(\cdot|\cdot, i)$) only on $\mathcal{D}_i$, in this paper, we are interested in data-sharing schemes that correspond to relabeling data from a different task, $j \neq i$ with the reward function $r_i$, and learn $\pi(\cdot|\cdot, i)$ on the combined data. To be able to do so, we assume access to the functional form of the reward $r_i$, a common assumption in goal-conditioned RL [3, 16], and which often holds in robotics applications through the use of learned classifiers [83, 32], and discriminators [18, 9].

We assume that relabeling data $\mathcal{D}_j$ from task $j$ to task $i$ generates a dataset $\mathcal{D}_{j \to i}$, which is then additionally used to train on task $i$. Thus, the effective dataset for task $i$ after relabeling is given by $\mathcal{D}_i^{\text{eff}} := \mathcal{D}_i \cup (\cup_{j \neq i} \mathcal{D}_{j \to i})$. This notation simply formalizes data sharing and relabeling strategies

explored in prior work [16, 33]. Our aim in this paper will be to improve on this naïve strategy, which we will show leads to significantly better results.

**Offline RL algorithms.** A central challenge in offline RL is distributional shift: differences between the learned policy and the behavior policy can lead to erroneous target values, where the Q-function is queried at actions $\mathbf{a} \sim \pi(\mathbf{a}|\mathbf{s})$ that are far from the actions it is trained on, leading to massive overestimation [43, 37]. A number of offline RL algorithms use some kind of regularization on either the policy [37, 20, 82, 29, 68, 57] or on the learned Q-function [39, 36] to ensure that the learned policy does not deviate too far from the behavior policy. For our analysis in this work, we will abstract these algorithms into a generic constrained policy optimization problem [39]:

$$\pi^*(\mathbf{a}|\mathbf{s}) := \arg\max_{\pi} \ J_{\mathcal{D}}(\pi) - \alpha D(\pi, \pi_{\beta}). \tag{1}$$

$J_{\mathcal{D}}(\pi)$ denotes the average return of policy $\pi$ in the empirical MDP induced by the transitions in the dataset, and $D(\pi, \pi_{\beta})$ denotes a divergence measure (e.g., KL-divergence [29, 82], MMD distance [37] or $D_{\mathrm{CQL}}$ [39]) between the learned policy $\pi$ and the behavior policy $\pi_{\beta}$. In the multi-task offline RL setting with data-sharing, the generic optimization problem in Equation 1 for a task $i$ utilizes the effective dataset $\mathcal{D}_i^{\mathrm{eff}}$. In addition, we define $\pi_{\beta}^{\mathrm{eff}}(\mathbf{a}|\mathbf{s}, i)$ as the effective behavior policy for task $i$ and it is given by: $\pi_{\beta}^{\mathrm{eff}}(\mathbf{a}|\mathbf{s}, i) := |\mathcal{D}_i^{\mathrm{eff}}(\mathbf{s}, \mathbf{a})|/|\mathcal{D}_i^{\mathrm{eff}}(\mathbf{s})|$. Hence, the counterpart of Equation 1 in the multi-task offline RL setting with data sharing is given by:

$$\forall i \in [N], \ \ \pi^*(\mathbf{a}|\mathbf{s}, i) := \arg\max_{\pi} \ J_{\mathcal{D}_i^{\mathrm{eff}}}(\pi) - \alpha D(\pi, \pi_{\beta}^{\mathrm{eff}}). \tag{2}$$

We will utilize this generic optimization problem to motivate our method in Section 5.

## 4  When Does Data Sharing Actually Help in Offline Multi-Task RL?

Our goal is to leverage experience from all tasks to learn a policy for a particular task of interest. Perhaps the simplest approach to leveraging experience across tasks is to train the task policy on not just the data coming from that task, but also relabeled data from all other tasks [6]. Is this naïve data sharing strategy sufficient for learning effective behaviors from multi-task offline data? In this section, we aim to answer this question via empirical analysis on a relatively simple domain, which will reveal interesting aspects of data sharing. We first describe the experimental setup and then discuss the results and possible explanations for the observed behavior. Using insights obtained from this analysis, we will then derive a simple and effective data sharing strategy in Section 5.

**Experimental analysis setup.** To assess the efficacy of data sharing, we experimentally analyze various multi-task RL scenarios created with the walker2d environment in Gym [5]. We construct different test scenarios on this environment that mimic practical situations, including settings where different amounts of data of varied quality are available for different tasks [33, 84, 69]. In all these scenarios, the agent attempts three tasks: `run forward`, `run backward`, and `jump`, which we visualize in Figure 3. Following the problem statement in Section 3, these tasks share the same state-action space and transition dynamics, differing only in the reward function that the agent is trying to optimize. Different scenarios are generated with varying size offline datasets, each collected with policies that have different degrees of suboptimality. This might include, for each task, a single policy with mediocre or expert performance, or a mixture of policies given by the initial part of the replay buffer trained with online SAC [26]. We refer to these three types of offline datasets as medium, expert and medium-replay, respectively, following Fu et al. [19].

We train a single-task policy $\pi_{\mathrm{CQL}}(\mathbf{a}|\mathbf{s}, i)$ with CQL [39] as the base offline RL method, along with two forms of data-sharing, as shown in Table 1: no sharing of data across tasks (**No Sharing**)) and complete sharing of data with relabeling across all tasks (**Sharing All**). In addition, we also measure the divergence term in Equation 2, $D(\pi(\cdot|\cdot, i), \pi_{\beta}^{\mathrm{eff}}(\cdot|\cdot, i))$, for $\pi = \pi_{\mathrm{CQL}}(\mathbf{a}|\mathbf{s}, i)$, averaged across tasks by using the Kullback-Liebler divergence. This value quantifies the average divergence between the single-task optimal policy and the relabeled behavior policy averaged across tasks.

**Analysis of results in Table 1.** To begin, note that even naïvely sharing data is better than not sharing any data at all on **5/9** tasks considered (compare the performance across **No Sharing** and **Sharing All** in Table 1). However, a closer look at Table 1 suggests that data-sharing can significantly degrade performance on certain tasks, especially in scenarios where the amount of data available for the original task is limited, and where the distribution of this data is narrow. For example, when using

| Dataset types / Tasks | Dataset Size | Avg Return | | $D_{\text{KL}}(\pi, \pi_\beta)$ | |
|---|---|---|---|---|---|
| | | No Sharing | Sharing All | No Sharing | Sharing All |
| medium-replay / `run forward` | 109900 | **998.9** | 966.2 | **3.70** | 10.39 |
| medium-replay / `run backward` | 109980 | **1298.6** | 1147.5 | **4.55** | 12.70 |
| medium-replay / `jump` | 109511 | **1603.1** | 1224.7 | **3.57** | 15.89 |
| **average task performance** | N/A | **1300.2** | 1112.8 | **3.94** | 12.99 |
| medium / `run forward` | 27646 | 297.4 | **848.7** | **6.53** | 11.78 |
| medium / `run backward` | 31298 | 207.5 | **600.4** | **4.44** | 10.13 |
| medium / `jump` | 100000 | 351.1 | **776.1** | **5.57** | 21.27 |
| **average task performance** | N/A | 285.3 | **747.7** | **5.51** | 14.39 |
| medium-replay / `run forward` | 109900 | 590.1 | **701.4** | **1.49** | 7.76 |
| medium / `run backward` | 31298 | 614.7 | **756.7** | **1.91** | 12.2 |
| expert / `jump` | 5000 | **1575.2** | 885.1 | **3.12** | 27.5 |
| **average task performance** | N/A | **926.6** | 781 | **2.17** | 15.82 |

Table 1: We analyze how sharing data across all tasks (**Sharing All**) compares to **No Sharing** in the multi-task walker2d environment with three tasks: run forward, run backward, and jump. We provide three scenarios with different styles of per-task offline datasets in the leftmost column. The second column shows the number of transitions in each dataset. We report the per-task average return, the KL divergence between the single-task optimal policy $\pi$ and the behavior policy $\pi_\beta$ after the data sharing scheme, as well as averages across tasks. **Sharing All** generally helps training while increasing the KL divergence. However, on the row highlighted in yellow, **Sharing All** yields a particularly large KL divergence between the single-task $\pi$ and $\pi_\beta$ and degrades the performance, suggesting sharing data for all tasks is brittle.

expert data for jumping in conjunction with more than 25 times as much lower-quality (mediocre & random) data for running forward and backward, we find that the agent performs poorly on the jumping task despite access to near-optimal jumping data.

***Why does naïve data sharing degrade performance on certain tasks despite near-optimal behavior for these tasks in the original task dataset?*** We argue that the primary reason that naïve data sharing can actually hurt performance in such cases is because it exacerbates the distributional shift issues that afflict offline RL. Many offline RL methods combat distribution shift by implicitly or explicitly constraining the learned policy to stay close to the training data. Then, when the training data is changed by adding relabeled data from another task, the constraint causes the learned policy to change as well. When the added data is of low quality for that task, it will correspondingly lead to a lower quality learned policy for that task, unless the constraint is somehow modified. This effect is evident from the higher divergence values between the learned policy without any data-sharing and the effective behavior policy for that task *after* relabeling (e.g., expert+`jump`) in Table 1. Although these results are only for CQL, we expect that any offline RL method would, insofar as it combats distributional shift by staying close to the data, would exhibit a similar problem.

**To mathematically quantify** the effects of data-sharing in multi-task offline RL, we appeal to safe policy improvement bounds [41, 39, 92] and discuss cases where data-sharing between tasks $i$ and $j$ can degrade the amount of worst-case guaranteed improvement over the behavior policy. Prior work [39] has shown that the generic offline RL algorithm in Equation 1 enjoys the following guarantees of policy improvement on the actual MDP, beyond the behavior policy:

$$J(\pi^*) \geq J(\pi_\beta) - \mathcal{O}(1/(1-\gamma)^2)\mathbb{E}_{\mathbf{s},\mathbf{a}\sim d^\pi}\left[\sqrt{\frac{D(\pi(\cdot|\mathbf{s}), \pi_\beta(\cdot|\mathbf{s}))}{|\mathcal{D}(\mathbf{s})|}}\right] + \alpha/(1-\gamma)D(\pi, \pi_\beta). \quad (3)$$

We will use Equation 3 to understand the scenarios where data sharing can hurt. When data sharing modifies $\mathcal{D} = \mathcal{D}_i$ to $\mathcal{D} = \mathcal{D}_i^{\text{eff}}$, which includes $\mathcal{D}_i$ as a subset, it effectively aims at reducing the magnitude of the second term (i.e., sampling error) by increasing the denominator. This can be highly effective if the state distribution of the learned policy $\pi^*$ and the dataset $\mathcal{D}$ overlap. However, an increase in the divergence $D(\pi(\cdot|\mathbf{s}), \pi^\beta(\cdot|\mathbf{s}))$ as a consequence of relabeling implies a potential increase in the sampling error, unless the increased value of $|\mathcal{D}^{\text{eff}}(\mathbf{s})|$ compensates for this. Additionally, the bound also depends on the quality of the behavior data added after relabeling: if the resulting behavior policy $\pi_\beta^{\text{eff}}$ is more suboptimal compared to $\pi_\beta$, i.e., $J(\pi_\beta^{\text{eff}}) < J(\pi_\beta)$, then the guaranteed amount of improvement also reduces.

**To conclude,** our analysis reveals that while data sharing is often helpful in multi-task offline RL, it can lead to substantially poor performance on certain tasks as a result of exacerbated distributional shift between the optimal policy and the effective behavior policy induced after sharing data.

# 5 CDS: Reducing Distributional Shift in Multi-Task Data Sharing

The analysis in Section 4 shows that naïve data sharing may be highly sub-optimal in some cases, and although it often does improve over no data sharing at all in practice, it can also lead to exceedingly poor performance. Can we devise a conservative approach that shares data intelligently to not exacerbate distributional shift as a result of relabeling?

## 5.1 A First Attempt at Designing a Data Sharing Strategy

A straightforward data sharing strategy is to utilize a transition for training only if it reduces the distributional shift. Formally, this means that for a given transition $(\mathbf{s}, \mathbf{a}, r_j(\mathbf{s}, \mathbf{a}), \mathbf{s}') \in \mathcal{D}_j$ sampled from the dataset $\mathcal{D}_j$, such a scheme would prescribe using it for training task $i$ (i.e., $(\mathbf{s}, \mathbf{a}, r_i(\mathbf{s}, \mathbf{a}), \mathbf{s}') \in \mathcal{D}_i^{\text{eff}}$) only if:

$$\textbf{CDS (basic)}: \quad \Delta^{\pi}(\mathbf{s}, \mathbf{a}) := D(\pi(\cdot|\cdot, i), \pi_{\beta}(\cdot|\cdot, i))(\mathbf{s}) - D(\pi(\cdot|\cdot, i), \pi_{\beta}^{\text{eff}}(\cdot|\cdot, i))(\mathbf{s}) \geq 0. \quad (4)$$

The scheme presented in Equation 4 would guarantee that distributional shift (i.e., second term in Equation 2) is reduced. Moreover, since sharing data can only increase the size of the dataset and not reduce it, this scheme is guaranteed to not increase the sampling error term in Equation 3. We refer to this scheme as the basic variant of conservative data sharing (**CDS (basic)**).

While this scheme can prevent the negative effects of increased distributional shift, this scheme is quite pessimistic. Even in our experiments, we find that this variant of CDS does not improve performance by a large margin. Additionally, as observed in Table 1 (medium-medium-medium data composition) and discussed in Section 4, data sharing can often be useful despite an increased distributional shift (note the higher values of $D_{\text{KL}}(\pi, \pi_{\beta})$ in Table 1) likely because it reduces sampling error and potentially utilizes data of higher quality for training. **CDS (basic)** described above does not take into account these factors. Formally, the effect of the first term in Equation 2, $J_{\mathcal{D}^{\text{eff}}}(\pi)$ (the policy return in the empirical MDP generated by the dataset) and a larger increase in $|\mathcal{D}^{\text{eff}}(\mathbf{s})|$ at the cost of somewhat increased value of $D(\pi(\cdot|\mathbf{s}), \pi_{\beta}(\cdot|\mathbf{s})$ are not taken into account. Thus we ask: can we instead design a more complete version of CDS that effectively balances the tradeoff by incorporating all the discussed factors (distributional shift, sampling error, data quality)?

## 5.2 The Complete Version of Conservative Data Sharing (CDS)

Next, we present the complete version of our method. The complete version of CDS, which we will refer to as **CDS**, for notational brevity is derived from the following perspective: we note that a data sharing scheme can be viewed as altering the dataset $\mathcal{D}_i^{\text{eff}}$, and hence the effective behavior policy, $\pi_{\beta}^{\text{eff}}(\mathbf{a}|\mathbf{s}, i)$. Thus, we can directly *optimize* the objective in Equation 2 with respect to $\pi_{\beta}^{\text{eff}}$, in addition to $\pi$, where $\pi_{\beta}^{\text{eff}}$ belongs to the set of all possible effective behavior policies that can be obtained via any form of data sharing. Note that unlike CDS (basic), this

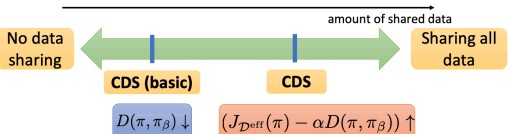

Figure 2: A schematic comparing **CDS** and **CDS (basic)** data sharing schemes relative to no sharing (left extreme) and full data sharing (right extreme). While p-CDS only shares data when distributional shift is strictly reduced, o-CDS is more optimistic and shares data when the objective in Equation 2 is larger. Typically, we would expect that CDS shares more transitions than CDS (basic).

approach would not rely on only indirectly controlling the objective in Equation 2 by controlling distributional shift, but would aim to directly optimize the objective in Equation 2. We formalize this optimization below in Equation 5:

$$\arg\max_{\pi} \max_{\pi_{\beta}^{\text{eff}} \in \Pi_{\text{relabel}}} \left[ J_{\mathcal{D}_i^{\text{eff}}}(\pi) - \alpha D(\pi, \pi_{\beta}^{\text{eff}}; i) \right], \quad (5)$$

where $\Pi_{\text{relabel}}$ denotes the set of all possible behavior policies that can be obtained via relabeling. The next result characterizes safe policy improvement for Equation 5 and discusses how it leads to improvement over the behavior policy and also produces an effective practical method.

**Proposition 5.1** (Characterizing safe-policy improvement for CDS.). *Let $\pi^*(\mathbf{a}|\mathbf{s})$ be the policy obtained by optimizing Equation 5, and let $\pi_{\beta}(\mathbf{a}|\mathbf{s})$ be the behavior policy for $\mathcal{D}_i$. Then, w.h.p.*

$\geq 1 - \delta$, $\pi^*$ is a $\zeta$-safe policy improvement over $\pi_\beta$, i.e., $J(\pi^*) \geq J(\pi_\beta) - \zeta$, where $\zeta$ is given by:

$$\zeta = \mathcal{O}\left(\frac{1}{(1-\gamma)^2}\right) \mathbb{E}_{\mathbf{s} \sim d^{\pi^*}_{\mathcal{D}^{\mathrm{eff}}_i}} \left[\sqrt{\frac{D_{CQL}(\pi^*, \pi^*_\beta)(\mathbf{s}) + 1}{|\mathcal{D}^{\mathrm{eff}}_i(\mathbf{s})|}}\right] - \left[\alpha D(\pi^*, \pi^*_\beta) + \underbrace{J(\pi^*_\beta) - J(\pi_\beta)}_{(a)}\right],$$

where $\mathcal{D}^{\mathrm{eff}}_i \sim d^{\pi^*_\beta}(\mathbf{s})$ and $\pi^*_\beta(\mathbf{a}|\mathbf{s})$ denotes the policy $\pi \in \Pi_{relabel}$ that maximizes Equation 5.

A proof and analysis of this proposition is provided in Appendix B, where we note that the bound in Proposition 5.1 is stronger than both no data sharing as well as naïve data sharing. We show in Appendix B that optimizing Equation 5 reduces the numerator $D_{\mathrm{CQL}}(\pi^*, \pi^*_\beta)$ term while also increasing $|\mathcal{D}^{\mathrm{eff}}_i(\mathbf{s})|$, thus reducing the amount of sampling error. In addition, Lemma B.1 shows that the improvement term $(a)$ is guaranteed to be positive if a large enough $\alpha$ is chosen in Equation 5. Combining these, we find data sharing using Equation 5 improves over both complete data sharing (which may increase $D_{\mathrm{CQL}}(\pi, \pi_\beta)$) and no data sharing (which does not increase $|\mathcal{D}^{\mathrm{eff}}_i(\mathbf{s})|$). A schematic comparing the two variants of CDS and naïve and no data sharing schemes is shown in Figure 2.

**Optimizing Equation 5 tractably.** The next step is to effectively convert Equation 5 into a simple condition for data sharing in multi-task offline RL. While directly solving Equation 5 is intractable in practice, since both the terms depend on $\pi^{\mathrm{eff}}_\beta(\mathbf{a}|\mathbf{s})$ (since the first term $J_{\mathcal{D}^{\mathrm{eff}}}(\pi)$ depends on the empirical MDP induced by the effective behavior policy and the amount of sampling error), we need to instead solve Equation 5 approximately. Fortunately, we can optimize a *lower-bound approximation* to Equation 5 that uses the dataset state distribution for the policy update in Equation 5 similar to modern actor-critic methods [12, 46, 21, 26, 39] which only introduces an additional $D(\pi, \pi_\beta)$ term in the objective. This objective is given by: $\mathbb{E}_{\mathbf{s} \sim \mathcal{D}^{\mathrm{eff}}_i}[\mathbb{E}_\pi[Q(\mathbf{s}, \mathbf{a}, i)] - \alpha' D(\pi(\cdot|\mathbf{s}, i), \pi^{\mathrm{eff}}_\beta(\cdot|\mathbf{s}, i))]$, which is equal to the expected "conservative Q-value" $\hat{Q}^\pi(\mathbf{s}, \mathbf{a}, i)$ on dataset states, policy actions and task $i$. Optimizing this objective via a co-ordinate descent on $\pi$ and $\pi^{\mathrm{eff}}_\beta$ dictates that $\pi$ be updated using a standard update of maximizing the conservative Q-function, $\hat{Q}^\pi$ (equal to the difference of the Q-function and $D(\pi, \pi^{\mathrm{eff}}_\beta; i)$). Moreover, $\pi^{\mathrm{eff}}_\beta$ should also be updated towards maximizing the same expectation, $\mathbb{E}_{\mathbf{s}, \mathbf{a} \sim \mathcal{D}^{\mathrm{eff}}_i}[\hat{Q}^\pi(\mathbf{s}, \mathbf{a}, i)] := \mathbb{E}_{\mathbf{s}, \mathbf{a} \sim \mathcal{D}^{\mathrm{eff}}_i}[Q(\mathbf{s}, \mathbf{a}, i)] - \alpha D(\pi, \pi^{\mathrm{eff}}_\beta; i)$. This implies that when updating the behavior policy during relabeling, we should prefer state-action pairs that maximize the conservative Q-function.

**Deriving the data sharing strategy for CDS.** Utilizing the insights for optimizing Equation 5 tractably as discussed above, we now present the effective data sharing rule prescribed by CDS. For any given task $i$, we want relabeling to incorporate transitions with the highest conservative Q-value into the resulting dataset $\mathcal{D}^{\mathrm{eff}}_i$, as this will directly optimize the tractable lower bound on Equation 5. While directly optimizing Equation 5 will enjoy benefits of reduced sampling error since $J_{\mathcal{D}^{\mathrm{eff}}_i}(\pi)$ also depends on sampling error, our tractable lower bound approximation does not enjoy this benefit. This is because optimizing the lower-bound only increases the frequency of a state in the dataset, $|\mathcal{D}^{\mathrm{eff}}_i(\mathbf{s})|$ by almost 1. To encourage further reduction in sampling error, we modify CDS to instead share all transitions with a conservative Q-value more than the top $k^{\mathrm{th}}$ quantile of the original dataset $\mathcal{D}_i$, where $k$ is a hyperparameter. This provably increases the objective value in Equation 5 still ensuring that term $(a) > 0$ in Proposition 5.1, while also reducing $|\mathcal{D}^{\mathrm{eff}}_i(\mathbf{s})|$ in the denominator. Thus, for a given transition $(\mathbf{s}, \mathbf{a}, \mathbf{s}') \in \mathcal{D}_j$,

> **CDS:** $(\mathbf{s}, \mathbf{a}, r_i, \mathbf{s}') \in \mathcal{D}^{\mathrm{eff}}_i$ if $\Delta^\pi(\mathbf{s}, \mathbf{a}) := \hat{Q}^\pi(\mathbf{s}, \mathbf{a}, i) - P_{k\%}\left\{\hat{Q}^\pi(\mathbf{s}', \mathbf{a}', i) : \mathbf{s}', \mathbf{a}' \sim \mathcal{D}_i\right\} \geq 0,$ (6)

where $\hat{Q}^\pi$ denotes the learned conservative Q-function estimate. If the condition in Equation 6 holds for the given $(\mathbf{s}, \mathbf{a})$, then the corresponding relabeled transition, $(\mathbf{s}, \mathbf{a}, r_i(\mathbf{s}, \mathbf{a}), \mathbf{s}')$ is added to $\mathcal{D}^{\mathrm{eff}}_i$.

We summarize the pesudocode of CDS in Algorithm 1 in Appendix A and include the practical implementation details of CDS in Appendix C.

# 6  Experimental Evaluation

We conduct experiments to answer six main questions: **(1)** can CDS prevent performance degradation when sharing data as observed in Section 4?, **(2)** how does CDS compare to vanilla multi-task offline RL methods and prior data sharing methods? **(3)** can CDS handle sparse reward settings, where data sharing is particularly important due to scarce supervision signal? **(4)** can CDS handle goal-conditioned offline RL settings where the offline dataset is undirected and highly suboptimal? **(5)** Can CDS scale to complex visual observations? **(6)** Can CDS be combined with any offline RL algorithms? Besides these questions, we visualize CDS weights for better interpretation of the data sharing scheme learned by CDS in Figure 4 in Appendix D.2.

**Comparisons.** To answer these questions, we consider the following prior methods. On tasks with low dimensional state spaces, we compare with the online multi-task relabeling approach **HIPI** [16], which uses inverse RL to infer for which tasks the datapoints are optimal and in practice routes a transition to task with the highest Q-value. We adapt HIPI to the offline setting by applying its data routing strategy to a conservative offline RL algorithm. We also compare to naïvely sharing data across all

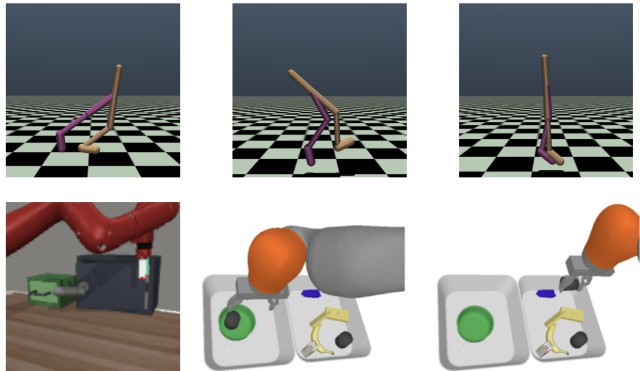

Figure 3: Environments (from left to right): walker2d run forward, walker2d run backward, walker2d jump, Meta-World door open/close and drawer open/close and vision-based pick-place tasks in [33].

tasks (denoted as **Sharing All**) and vanilla multi-task offline RL method without any data sharing (denoted as **No Sharing**). On image-based domains, we compare CDS to the data sharing strategy based on human-defined skills [33] (denoted as **Skill**), which manually groups tasks into different skills (e.g. skill "pick" and skill "place") and only routes an episode to target tasks that belongs to the same skill. In these domains, we also compare to **HIPI**, **Sharing All** and **No Sharing**. Beyond these multi-task RL approaches with data sharing, to assess the importance of data sharing in offline RL, we perform an additional comparison to other alternatives to data sharing in multi-task offline RL settings. One traditionally considered approach is to use data from other tasks for some form of "pre-training" before learning to solve the actual task. We instantiate this idea by considering a method from Yang and Nachum [86] that conducts contrastive representation learning on the multi-task datasets to extract shared representation between tasks and then runs multi-task RL on the learned representations. We discuss this comparison in detail in Table 7 in Appendix D.3. To answer question (6), we use CQL [39] (a Q-function regularization method) and BRAC [82] (a policy-constraint method) as the base offline RL algorithms for all methods. We discuss evaluations of CDS with CQL in the main text and include the results of CDS with BRAC in Table 5 in Appendix D.1. For more details on setup and hyperparameters, see Appendix C.

**Multi-task environments.** We consider a number of multi-task reinforcement learning problems on environments visualized in Figure 3. To answer questions (1) and (2), we consider the walker2d locomotion environment from OpenAI Gym [5] with dense rewards. We use three tasks, `run forward`, `run backward` and `jump`, as proposed in prior offline RL work [91]. To answer question (3), we also evaluate on robotic manipulation domains using environments from the Meta-World benchmark [90]. We consider four tasks: `door open`, `door close`, `drawer open` and `drawer close`. Meaningful data sharing requires a consistent state representation across tasks, so we put both the door and the drawer on the same table, as shown in Figure 3. Each task has a sparse reward of 1 when the success condition is met and 0 otherwise. To answer question (4), we consider maze navigation tasks where the temporal "stitching" ability of an offline RL algorithm is crucial to obtain good performance. We create goal reaching tasks using the ant robot in the medium and hard mazes from D4RL [19]. The set of goals is a fixed discrete set of size 7 and 3 for large and medium mazes, respectively. Following Fu et al. [19], a reward of +1 is given and the episode terminates if the state is within a threshold radius of the goal. Finally, to explore how CDS scales to image-based manipulation tasks (question (5)), we utilize a simulation environment similar to the real-world setup presented in [33]. This environment, which was utilized by Kalashnikov et al. [33]

as a representative and realistic simulation of a real-world robotic manipulation problem, consists of 10 image-based manipulation tasks that involve different combinations of picking specific objects (banana, bottle, sausage, milk box, food box, can and carrot) and placing them in one of the three fixtures (bowl, plate and divider plate) (see example task images in Fig. 3). More environment details are in the appendix. We report the average return for locomotion tasks and success rate for AntMaze and both manipluation environments, averaged over 6 and 3 random seeds for environments with low-dimensional inputs and image inputs respectively.

| Environment | Dataset types / Tasks | $D_{\mathrm{KL}}(\pi, \pi_\beta)$ | | | |
|---|---|---|---|---|---|
| | | No Sharing | Sharing All | CDS (basic) (ours) | CDS (ours) |
| walker2d | medium-replay / run forward | **1.49** | 7.76 | 14.31 | **1.49** |
| | medium / run backward | **1.91** | 12.2 | 8.26 | 6.09 |
| | expert / jump | 3.12 | 27.5 | 13.25 | **2.91** |

Table 2: Measuring $D_{\mathrm{KL}}(\pi, \pi_\beta)$ on the walker2d environment. **Sharing All** degrades the performance on task jump with limited expert data as discussed in Table 1. CDS manages to obtain a $\pi_\beta$ after data sharing that is closer to the single-task optimal policy in terms of the KL divergence compared to **No Sharing** and **Sharing All** on task jump (highlighted in yellow). Since CDS also achieves better performance, this analysis suggests that reducing distribution shift is important for effective offline data sharing.

**Multi-task datasets.** Following the analysis in Section 4, we intentionally construct datasets with a variety of heterogeneous behavior policies to test if CDS can provide effective data sharing to improve performance while avoiding harmful data sharing that exacerbates distributional shift. For the locomotion domain, we use a large, diverse dataset (medium-replay) for run forward, a medium-sized dataset for run backward, and an expert dataset with limited data for run jump. For Meta-World, we consider medium-replay datasets with 152K transitions for task door close and drawer open and expert datasets with only 2K transitions for task door open and drawer close. For AntMaze, we modify the D4RL datasets for antmaze-*-play environments to construct two kinds of multi-task datasets: an "undirected" dataset, where data is equally divided between different tasks and the rewards are correspondingly relabeled, and a "directed" dataset, where a trajectory is associated with the goal closest to the final state of the trajectory. This means that the per-task data in the undirected setting may not be relevant to reaching the goal of interest. Thus, data-sharing is crucial for good performance: methods that do not effectively perform data sharing and train on largely task-irrelevant data are expected to perform worse. Finally, for image-based manipulation tasks, we collect datasets for all the tasks individually by running online RL [32] until the task reaches medium-level performance (40% for picking tasks and 80% placing tasks). At that point, we merge the entire replay buffers from different tasks creating a final dataset of 100K RL episodes with 25 transitions for each episode.

**Results on domains with low-dimensional states.** We present the results on all non-vision environments in Table 3. CDS achieves the best average performance across all environments except that on walker2d, it achieves the second best performance, obtaining slightly worse the other variant CDS (basic). On the locomotion domain, we observe the most significant improvement on task jump on all three environments. We interpret this as strength of conservative data sharing, which mitigates the distribution shift that can be introduced by routing large amount of other task data to the task with limited data and narrow distribution. We also validate this by measuring the $D_{\mathrm{KL}}(\pi, \pi_\beta)$ in Table 2 where $\pi_\beta$ is the behavior policy after we perform CDS to share data. As shown in Table 2, CDS achieves lower KL divergence between the single-task optimal policy and the behavior policy after data sharing on task jump with limited expert data, whereas **Sharing All** results in much higher KL divergence compared to **No Sharing** as discussed in Section 4 and Table 1. Hence, CDS is able to mitigate distribution shift when sharing data and result in performance boost.

On the Meta-World tasks, we find that the agent without data sharing completely fails to solve most of the tasks due to the low quality of the medium replay datasets and the insufficient data for the expert datasets. **Sharing All** improves performance since in the sparse reward settings, data sharing can introduce more supervision signal and help training. CDS further improves over **Sharing All**, suggesting that CDS can not only prevent harmful data sharing, but also lead to more effective multi-task learning compared to **Sharing All** in scenarios where data sharing is imperative. It's worth noting that CDS (basic) performs worse than CDS and **Sharing All**, indicating that relabeling data that only mitigates distributional shift is too pessimistic and might not be sufficient to discover the shared structure across tasks.

In the AntMaze tasks, we observe that CDS performs better than **Sharing All** and significantly outperforms HIPI in all four settings. Perhaps surprisingly, **No Sharing** is a strong baseline, however,

| Environment | Tasks / Dataset type | CDS (ours) | CDS (basic) | HIPI [16] | Sharing All | No Sharing |
|---|---|---|---|---|---|---|
| walker2d | run forward / medium-replay | **1057.9**±121.6 | 968.6±188.6 | 695.5±61.9 | 701.4±47.0 | 590.1±48.6 |
| | run backward / medium | 564.8±47.7 | 594.5±22.7 | 626.0±48.0 | **756.7**±76.7 | 614.7±87.3 |
| | jump / expert | 1418.2±138.4 | 1501.8±115.1 | **1603.7**±146.8 | 885.1±152.9 | 1575.2±70.9 |
| | average | 1013.6±71.5 | **1021.6**±76.9 | 975.1±45.1 | 781.0±100.8 | 926.6±37.7 |
| Meta-World [90] | door open / expert | **58.4%**±9.3% | 30.1%±16.6% | 26.5%±20.5% | 34.3%±17.9% | 14.5%±12.7 |
| | door close / medium-replay | **65.3%**±27.7% | 41.5%±28.2% | 1.3%±5.3% | 48.3%±27.3% | 4.0%±6.1% |
| | drawer open / medium-replay | **57.9%**±16.2% | 39.4%±16.9% | 41.2%±24.9% | 55.1%±9.4% | 16.0%±17.5% |
| | drawer close / expert | 98.8%±0.7% | 86.3%±0.9% | 62.2%±33.4% | **100.0%**±0% | 99.0%±0.7% |
| | average | **70.1%**±8.1% | 49.3%±16.0% | 32.8%±18.7% | 59.4%±5.7% | 33.4%±8.3% |
| AntMaze [19] | large maze (7 tasks) / undirected | **22.8%** ± 4.5% | 10.0% ± 5.9% | 1.3% ± 2.3% | 16.7% ± 7.0% | 13.3% ± 8.6% |
| | large maze (7 tasks) / directed | **24.6%** ± 4.7% | 0.0% ± 0.0% | 11.8% ± 5.4% | 20.6% ± 4.4% | 19.2% ± 8.0% |
| | medium maze (3 tasks) / undirected | **36.7%** ± 6.2% | 0.0% ± 0.0% | 8.6% ± 3.2% | 22.9% ± 3.6% | 21.6% ± 7.1% |
| | medium maze (3 tasks) / directed | **18.5%** ± 6.0% | 0.0% ± 0.0% | 8.3% ± 9.1% | 12.4% ± 5.4% | **17.0%** ± 3.2% |

Table 3: Results for multi-task locomotion (walker2d), robotic manipulation (Meta-World) and navigation environments (AntMaze) with low-dimensional state inputs. Numbers are averaged across 6 seeds, ± the 95%-confidence interval. We include per-task performance for walker2d and Meta-World domains and the overall performance averaged across tasks (highlighted in gray) for all three domains. We bold the highest score across all methods. CDS achieves the best or comparable performance on all of these environments.

| Task Name | CDS (ours) | HIPI [16] | Skill [33] | Sharing All | No Sharing |
|---|---|---|---|---|---|
| lift-banana | **53.1%**±3.2% | 48.3%±6.0% | 32.1%±9.5% | 41.8%±4.2% | 20.0%±6.0% |
| lift-bottle | **74.0%**±6.3% | 64.4%±7.7% | 55.9%±9.6% | 60.1%±10.2% | 49.7%±8.7% |
| lift-sausage | **71.8%**±3.9% | 71.0%±7.7% | 68.8%±9.3% | 70.0%±7.0% | 60.9%±6.6% |
| lift-milk | **83.4%**±5.2% | 79.0%±3.9% | 68.2%±3.5% | 72.5%±5.3% | 68.4%±6.1% |
| lift-food | 61.4%±9.5% | **62.6%**±6.3% | 41.5%±12.1% | 58.5%±7.0% | 39.1%±7.0% |
| lift-can | 65.5%±6.9% | **67.8%**±6.8% | 50.8%±12.5% | 57.7%±7.2% | 49.1%±9.8% |
| lift-carrot | **83.8%**±3.5% | 78.8%±6.9% | 66.0%±7.0% | 75.2%±7.6% | 69.4%±7.6% |
| place-bowl | **81.0%**±8.1% | 77.2%±8.9% | 80.8%±6.9% | 70.8%±7.8% | 80.3%±8.6% |
| place-plate | 85.8%±6.6% | 83.6%±7.9% | 78.4%±9.6% | 78.7%±7.6% | **86.1%**±7.7% |
| place-divider-plate | **87.8%**±7.6% | 78.0%±10.5% | 80.8%±5.3% | 79.2%±6.3% | 85.0%±5.9% |
| **average** | **74.8%**±6.4% | 71.1%±7.5% | 62.3%±8.9% | 66.4%±7.2% | 60.8%±7.5% |

Table 4: Results for multi-task vision-based robotic manipulation domains in [33]. Numbers are averaged across 3 seeds, ± the 95% confidence interval. We consider 7 tasks denoted as lift-object where the goal of each task is to lift a different object and 3 tasks denoted as place-fixture that aim to place a lifted object onto different fixtures. CDS outperforms both a skill-based data sharing strategy [33] (**Skill**) and other data sharing methods on the average task success rate (highlighted in gray) and 7 out of 10 per-task success rates.

is outperformed by CDS with the harder undirected data. Moreover, CDS performs on-par or better in the undirected setting compared to the directed setting, indicating the effectiveness of CDS in routing data in challenging settings.

**Results on image-based robotic manipulation domains.** Here, we compare CDS to the hand-designed **Skill** sharing strategy, in addition to the other methods. Given that CDS achieves significantly better performance than CDS (basic) on low-dimensional robotic manipulation tasks in Meta-World, we only evaluate CDS in the vision-based robotic manipulation domains. Since CDS is applicable to any offline multi-task RL algorithm, we employ it as a separate data-sharing strategy in [33] while keeping the model architecture and all the other hyperparameters constant, which allows us to carefully evaluate the influence of data sharing in isolation. The results are reported in Table 4. CDS outperforms both **Skill** and other approaches, indicating that CDS is able to scale to high-dimensional observation inputs and can effectively remove the need for manual curation of data sharing strategies.

## 7 Conclusion

In this paper, we study the multi-task offline RL setting, focusing on the problem of sharing offline data across tasks for better multi-task learning. Through empirical analysis, we identify that naïvely sharing data across tasks generally helps learning but can significantly hurt performance in scenarios where excessive distribution shift is introduced. To address this challenge, we present conservative data sharing (CDS), which relabels data to a task when the conservative Q-value of the given transition is better than the expected conservative Q-value of the target task. On multitask locomotion, manipulation, navigation, and vision-based manipulation domains, CDS consistently outperforms or achieves comparable performance to existing data sharing approaches. While CDS attains superior results, it is not able to handle data sharing in settings where dynamics vary across tasks and requires functional forms of rewards. We leave these as future work.

## Acknowledgements

We thank Kanishka Rao, Xinyang Geng, Avi Singh, other members of RAIL at UC Berkeley, IRIS at Stanford and Robotics at Google and anonymous reviewers for valuable and constructive feedback on an early version of this manuscript. This research was funded in part by Google, ONR grants N00014-20-1-2675 and N00014-21-1-2685, Intel Corporation and the DARPA Assured Autonomy Program. CF is a CIFAR Fellow in the Learning in Machines and Brains program.

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
