# OpenReview forum: "Conservative Data Sharing for Multi-Task Offline Reinforcement Learning"
_NeurIPS.cc/2021/Conference — NeurIPS 2021 Poster_

### Official Review · Reviewer_dWyg · 2021-07-11

**Rating:** 6
**Confidence:** 3

**Summary:**

The paper considers multi-task RL under the offline setting where the underlying system dynamics is the same but different task has a different reward function. Through experimental analysis, the authors argue that naive data sharing, while intuitive, can potentially degrade the performance due to distributional shift. To address the issue, this work proposes a simple, conservative data sharing scheme that only shares data among tasks that could potentially have a higher conservative Q-value estimate. Experimental results on several tasks validate that the scheme can often be effective.

**Limitations And Societal Impact:**

Yes

**Main Review:**

The paper is well organized with clear presentation and meaningful motivation/analysis on data sharing. This makes the paper easy to follow and overall sound. While I enjoyed reading it, I think the significance of its contribution might be limited. The intuition is straightforward and the results are exactly what one would expect. In a sense, naively augmenting data that are too far from the original task could increase the effect of distributional shift and hurt the performance. The current work hence uses the "conservative approaches", common in offline RL, to regularize the data to be shared. These being said, I think the paper does  articulate the arguments well and the resulting algorithm seems to be an effective approach based on the experiments.

Additional comments:
1. the presented tables seem to base on experiments from a single run. Given how noisy the results could be, few more repeated experiments are helpful in understanding the gaps among approaches.
2. the authors mention the limitation of CDS on requiring functional forms of rewards. In lines 144-145, it is stated that this is a common assumption in goal-conditioned RL (or otherwise, through the use of learned classifiers). Since the authors also mention [7] as the offline RL work for this setting, it seems reasonable to compare for goal-conditioned problem if appropriate.

**Time Spent Reviewing:**

2.5h

---

> ### Author Response · Authors · 2021-08-10
> **Author Response**
>
> Thank you for your feedback and for a positive assessment of the paper! We answer the specific concerns below.
>
> **Significance of the contribution is limited:** We are glad that the reviewer found that the paper presents a clear description of an idea which is intuitive to follow and understand. Regarding significance, we would like to note that to the best of our knowledge, this paper is one of the first works that analyze the reasons behind why full data sharing can perform poorly  in multi-task offline RL, and proposes an effective method to address the problem. We believe that the proposed algorithm will enable practitioners to address multi-task offline RL problems more effectively, and that the analysis framework will inform the design of future algorithms.
>
> **More seeds:** To address this concern, we have now run more seeds for each algorithm on the walker2d and Meta-World tasks (Table 3) and the robotic manipulation tasks from vision (Table 4) and present the aggregate numbers in terms of average task performance in the two tables below. These new results are averaged over 8 seeds for walker2d and Meta-World tasks and 4 seeds for the vision-based robotic manipulation tasks and $\pm$ shows the 95%-confidence interval. The results indicate that CDS still outperforms other methods in line with the results in the paper.  We will run more seeds for other domains as well and present the results in the revised paper.
>
> | Environment | CDS (ours)      | HIPI           | Sharing All     | No Sharing     |
> |-------------|-----------------|----------------|-----------------|----------------|
> | walker2d (average task)    | 1053.5$\pm$71.5 | 964.7$\pm$45.1 | 926.6$\pm$100.8 | 781.0$\pm$37.7 |
> | Meta-World (average task)  | 70.1%$\pm$8.1%    | 32.8%$\pm$18.7%  | 59.4%$\pm$5.7%    | 33.4%$\pm$8.3%   |
>
> | Environment                                                      | CDS (ours)   | HIPI         | Skill[33]    | Sharing All  | No Sharing   |
> |------------------------------------------------------------------|--------------|--------------|--------------|--------------|--------------|
> | multi-task vision-based robotic manipulation [33] (average task) | 77.7%$\pm$2.0% | 66.6%$\pm$3.2% | 58.2%$\pm$2.3% | 63.5%$\pm$1.0% | 57.8%$\pm$2.1% |
>
> **CDS on goal-conditioned problems:** We will aim to evaluate CDS on a goal-conditioned task with an infinite number of target goals for the final version of this paper. We would also like to clarify that the paper includes a goal-conditioned experiment with a finite number of target goals on the antmaze domains in Table 3. In this case, the method in [7] would closely resemble the Sharing All baseline, which CDS outperforms in Table 3. We will make this more clear in the paper.

---

### Official Review · Reviewer_ceUo · 2021-07-11

**Rating:** 6
**Confidence:** 4

**Summary:**

This paper addresses the problem of data sharing in multi-task offline RL, and show that naive data sharing scheme can be harmful for some tasks where data sharing could lead to severe distribution shift. To effectively utilize data-sharing in such tasks, this paper proposes a conservative data sharing (CDS), that re-labels data whose conservative Q-value is higher than the expected conservative Q-value in the target dataset.

**Limitations And Societal Impact:**

Addressed in the Section 7.

**Main Review:**

- Problem setup is a bit limited but important, the proposed idea is simple and interesting, and considering distribution shift for utilizing multi-task datasets seems very reasonable.

- I have a strong concern on the significance of the experimental results. First of all, it's not clear why the standard deviations are only reported in the appendix and bolds for similar performance are only made in the tables from appendix (also missing the results on visual domain). It seems that there are many tasks where there is a overlap if we consider standard deviations, and this is important for evaluating the significance of the experimental results. I think there should be more runs for making it possible to make a conclusion on the effectiveness of the proposed method. Also, reporting learning / evaluation curves would be more helpful for understanding the performance.

-  Curious about how conservative Q-function generalizes to unseen state-action pairs in different datasets. Are the estimated Q-values at such unseen regime reliable estimate for filtering out 'bad' transitions? As CQL only regularizes unseen actions, it could not handle distribution shift from unseen states. Is there supporting analysis or experiments to show that the proposed method works as intended, filtering out 'bad' or 'not relevant' state-action pairs whose 'ground-truth' Q-values are low? Also, how does the proposed method compare to the baseline of just randomly sampling transitions from datasets of other tasks?

- In that sense, more qualitative analysis like how much of the datasets are assigned to each task should be more helpful. For example, showing that transitions from the similar tasks are assigned to the target task but transitions from tasks which are not relevant are not assigned would be interesting.

- Problem setup of sharing data across multiple tasks seems very important and I like the observations in Table 1, but it's a bit limited in that this paper only deals with the setup of shared state/action space, and reward function is known (as authors already stated in the paper). In that sense, important but missing baseline (related work) should be utilizing other datasets for learning representations, as in [Yang'21].

[Yang'21] Representation Matters: Offline Pretraining for Sequential Decision Making, ICML 2021

**Time Spent Reviewing:**

8 Hours

---

> ### Author Response · Authors · 2021-08-10
> **Author Response**
>
> Thank you for your detailed comments! Based on your feedback, we have now increased the number of seeds in the empirical results; added diagnostic metrics and visualizations to understand the efficacy of CDS in filtering our irrelevant or bad state-action pairs from other tasks; and compared to the representation learning method from Yang et al. 2021. We detail these experiments below:
>
> **Significance of empirical results:** We apologize for adding the standard deviations only in the appendix, and have now added these in the main paper as well, along with bold for similar performing algorithms. In addition, we have now run more seeds for each algorithm, totaling to 8 seeds for each method in the multi-task walker and Meta-World domains (Table 3). We have added the final numbers in the paper, and here we summarize the results in terms of the average task performance ($\pm$ denotes the 95%-confidence interval of the various runs). As shown below, CDS still consistently outperforms other data sharing schemes by a significant margin.
>
> | Environment | CDS (ours)      | HIPI           | Sharing All     | No Sharing     |
> |-------------|-----------------|----------------|-----------------|----------------|
> | walker2d (average task)    | 1053.5$\pm$71.5 | 964.7$\pm$45.1 | 926.6$\pm$100.8 | 781.0$\pm$37.7 |
> | Meta-World (average task)  | 70.1%$\pm$8.1%    | 32.8%$\pm$18.7%  | 59.4%$\pm$5.7%    | 33.4%$\pm$8.3%   |
>
> We have also now added the performance averaged over 4 random seeds along with 95%-confidence intervals for the visual domain from Table 4, and we find that CDS still clearly outperforms other methods as shown below. We will update these results in the final version of the paper.
>
> | Environment                                                      | CDS (ours)   | HIPI         | Skill[33]    | Sharing All  | No Sharing   |
> |------------------------------------------------------------------|--------------|--------------|--------------|--------------|--------------|
> | multi-task vision-based robotic manipulation [33] (average task) | 77.7%$\pm$2.0% | 66.6%$\pm$3.2% | 58.2%$\pm$2.3% | 63.5%$\pm$1.0% | 57.8%$\pm$2.1% |
>
> ___
>
> **Do estimated conservative Q-values effectively filter bad transitions? Transitions from similar tasks are assigned to target task but not irrelevant transitions?** To address these comments, we have now added some diagnostic metrics to understand the effectiveness of CDS in filtering out irrelevant/bad transitions from other tasks. We summarize the new results below:
>
> 1. On the Meta-World environment, we would expect that for a given target task, say Drawer Close, transitions from a task that involves a different object (door) and a different skill (open) would not be as useful for learning. To understand if CDS weights reflect this expectation, we compare the average CDS weights on transitions from all the other tasks to two target tasks, Door Open and Drawer Close, respectively. We sort the CDS weights in the descending order. As shown, indeed CDS assigns higher weights to more related tasks and thus shares data from those tasks. In particular, the CDS weights for relabeling data from the task that handles the same object as the target task are much higher than the weights for tasks that consider a different object. For example, when relabeling to the target task Door Open, datapoints from task Door Close are assigned with much higher weights than those from either task Drawer Open or task Drawer Close. This suggests that CDS filters the irrelevant transitions for learning a given task.
>
> | Relabeling direction         | CDS weight |
> |------------------------------|------------|
> | Door Close -> Door Open      | 0.46       |
> | Drawer Open -> Door Open     | 0.10       |
> | Drawer Close -> Door Open    | 0.02       |
> |------------------------------|------------|
> | Drawer Open -> Drawer Close  | 0.35       |
> | Door Open -> Drawer Close    | 0.26       |
> | Door Close -> Drawer Close   | 0.22       |
>
> 2. On the AntMaze-large environment with undirected data, we visualize the CDS weight for the various tasks (goals) in the form of a heatmap and present the results at this [anonymized URL](https://drive.google.com/file/d/1PXuzfTfU1MJeI3RX_B5mB3CRVD-hf2wZ/view). To generate this plot, we sample a set of state-action pairs from the entire dataset for all tasks, and then plot the weights assigned by CDS as the color of the point marker at the (x, y) locations of these state-action pairs in the maze. Each plot computes the CDS weight corresponding to the target task (goal) indicated by the red $\times$ in the plot. As can be seen, CDS assigns higher weights to transitions from nearby goals as compared to transitions from farther away goals. This matches our expectation: transitions from nearby (x, y) locations are likely to be the most useful in learning a particular target task and CDS chooses to share these transitions to the target task.
>
>
> 3. Finally, as already shown in the walker2d environment in Table 3, adding transitions dictated by CDS actually reduces the KL-divergence between the optimal policy for the jump task and the behavior policy from 3.12 (No sharing) to 2.91 (CDS). Since the only way this KL divergence can reduce compared to no-sharing is if CDS identifies and shares transitions from other tasks that are closer to the optimal policy of the target jumping task, this reduction in the KL divergence again indicates that CDS effectively shares good transitions that are actually relevant for solving the target task.
>
> ___
>
> **How does the proposed method compare to the baseline of just randomly sampling transitions from datasets of other tasks?** We compared CDS to a baseline that randomly samples tasks. To implement this baseline, for each task, we randomly sample transitions from other tasks and relabel them to the current task every epoch (1 epoch = 1000 gradient update steps on the critic and the actor). We evaluate this baseline on the Meta-World domain from Table 3. We denote this baseline as Random Sharing We show the results as follows (results averaged over 8 random seeds, $\pm$ the 95%-confidence interval).
>
> | Environment | CDS (ours)      | HIPI           | Sharing All     | No Sharing     | Random Sharing |
> |-------------|-----------------|----------------|-----------------|----------------|------------------|
> | Meta-World (average task)  | 70.1%$\pm$8.1%   | 32.8%$\pm$18.7%  | 59.4%$\pm$5.7%    | 33.4%$\pm$8.3%   | 59.9%$\pm$3.9%    |
>
> Observe that CDS still outperforms this baseline by a significant margin and this baseline performs similarly to Sharing All.
>
> ___
>
> **Missing representation learning baseline from Yang et al. 2021:** To address this comment, we performed an experiment on the Meta-World domain that first utilizes the data from all the tasks to learn a shared representation using the best method, ACL, from Yang et al. 2021 and then runs standard offline multi-task RL on top of this representation. We include the average task success rates of all tasks in the table below. While the representation learning approach improves over standard multi-task RL without representation learning (No Sharing) consistent with the findings in Yang et al 2021, we still find that CDS with no representation learning outperforms this representation learning approach by a large margin on multi-task performance.
>
> | Environment | CDS (ours)      | HIPI           | Sharing All     | No Sharing     | Yang et al. 2021 |
> |-------------|-----------------|----------------|-----------------|----------------|------------------|
> | Meta-World (average task)  | 70.1%$\pm$8.1%    | 32.8%$\pm$18.7%  | 59.4%$\pm$5.7%    | 33.4%$\pm$8.3%   | 38.7%$\pm$11.1%    |
>
> Finally, we also note that any representation learning approach is complementary to our data sharing strategy and CDS can be used for offline multi-task RL on top of any state (or state-action) representation. We will evaluate this combination for the final version of the paper.

---

> > ### Comment · Reviewer_ceUo · 2021-08-11
> > **Thanks for the response**
> >
> > Thank you very much for the response. I also have read the other reviews and find the responses make sense, and I believe incorporating additional experimental results, analysis, and discussion into the final draft can significantly improve the quality of the paper. I will raise my score from 5 to 6, but I have one remaining question, so it would be very nice authors could elaborate more on this even though I raise my score:
> >
> > - I really like the qualitative analysis in the respose, and it seems that the proposed method works well as intended. But one unresolved question is about 'how' Q-function could generalize, or extrapolate to 'states' from unseen tasks, as it seems very surprising to me. I think more plausible explanation or discussion on this could significantly make the contribution of this paper clear.

---

> > > ### Author Response · Authors · 2021-08-12
> > > **Thank you for your reply and Clarification**
> > >
> > > Thank you for your positive feedback and for updating your score! Regarding your question, the algorithm softly shares all data across all of the tasks as discussed in L284-288. Thus, by the end of training, data from all tasks has been seen by the parameters of the network. This means that while CDS downweights datapoints that are irrelevant to the task of interest using soft weights (Equation 6), as confirmed in the diagnostic visualizations, no individual datapoint is _completely_ unseen nor out-of-distribution. We will revise the paper to include a discussion of this point. Please let us know if you have any more questions.

---

### Official Review · Reviewer_Axin · 2021-07-14

**Rating:** 8
**Confidence:** 3

**Summary:**

Summary

In this paper, the authors study the problem of offline RL from a different angle: how do we pick the appropriate data samples from certain offline tasks so that the distribution shift between source tasks and target tasks is minimized.
Conservative data sharing (CDS) is proposed, which proposes a criteria of selecting the data samples from a theoretical inspiration.

**Ethical Concerns:**

No potential ethical concerns found.


**Limitations And Societal Impact:**

No potential negative societal impact found.
The limitations are addressed in the paper

**Main Review:**


Main Review

Pros:
1. Very good paper writing
The paper provides a comprehensive and exhaustive related work and background section.
It would be easy for readers to catch up the flow and understand the algorithm

2. Good mathematical support
The algorithm comes from a well supported mathematical ground.
The intuition is very natural, and the algorithm itself boils down to a simple and neat practical algorithm. It would provide valuable insight for future research

3. good experiment results
The experiments cover a sufficient spectrum, which includes experiments from low dimensional toy experiments to high dimensional simulated robotics environments.
And the baselines are adequate, showing the proposed algorithms improving upon previous work.

4. The algorithm is novel, and the topic of offline reinforcement learning is quite interesting and valuable to many real life applications.

5. code is released.

In general I am no expert in this area, but I think the quality of the paper is good.
However, for the experiments, the authors didn’t experiment with real life robotics tasks.
Considering the importance of offline RL for real life applications, these experiments will further improve the experiment section.


**Time Spent Reviewing:**

2 hours

---

> ### Author Response · Authors · 2021-08-10
> **Author Response**
>
> Thank you for your comments and a positive assessment of the paper! We are glad that you found the paper to be well written and supported by mathematical arguments and good empirical results.
>
> Per your suggestion of real robotics tasks, we have started to set up some real robotics tasks based on picking and placing multiple objects. We will add the results to the final version once the experiments are fully set-up and completed.

---

### Official Review · Reviewer_BHsT · 2021-07-14

**Rating:** 7
**Confidence:** 4

**Summary:**

This paper proposes a data-sharing strategy called CDS to tackle the distributional shift of the offline RL algorithm in the multitask setting. The authors reveal the problem of naive sharing strategies experimentally and theoretically. CDS achieves an overall advantage against naive sharing strategies in various multitask offline environments.

**Limitations And Societal Impact:**

The authors have addressed all of them.

**Main Review:**

$\textbf{Originality}$: This paper targets the multitask offline RL problem which has not been well studied, as far as I know, this is the first work to introduce the idea of data sharing into offline RL. Although the method proposed in this paper is still based on relabelling, the authors give a clear theoretical support for CDS under offline RL setting.

$\textbf{Quality}$: The quality of this paper is generally very good, it has solid theoretical and experimental support for its claims. The authors make a comprehensive evaluation on the proposed method under different settings, comparing with strong baselines.

$\textbf{Clarity}$: This paper is well-written but I have several comments. In Equation (4), why the argmax output is a tuple, I think $\pi_{\beta}^{\text{eff}}$ should not appear on the left side. In Table 2, what are the meaning of each number in the results column. In Figure 4, it's kind of weird to leave out the task image for AntMaze.

$\textbf{Significance}$: The technique proposed is simple and shown to be effective, and the problem addressed is meaningful. However, I have a concern why CDS is solely based on CQL, and the authors make the claim: "Although these results are only for CQL, we expect that any offline RL method would, insofar as it combats distributional shift by staying close to the data, would exhibit a similar problem". This is not so straightforward to me, and it'll be better if the authors could demonstrate the similar results (CDS, no sharing and sharing all) using another base offline RL algorithm.



**Time Spent Reviewing:**

5

---

> ### Author Response · Authors · 2021-08-10
> **Author Response**
>
> Thank you for your comments and for a positive assessment of our work! We respond to the queries below:
>
> **Applying CDS on a different base offline RL algorithm:** To address the reviewer’s concern, we implemented CDS on top of BRAC (Wu et al. 2019), a policy-constraint based offline RL method, which is different from CQL that penalizes Q-functions. As discussed in Section 5, we need to compute a conservative estimate of Q-values for CDS. While the Q-function from CQL directly provides us with this conservative estimate, BRAC does not directly learn a conservative Q-function estimator. Therefore,  for BRAC, we compute this conservative estimate by explicitly subtracting KL divergence between the learned policy $\pi(a|s)$ and the behavior policy $\pi^\beta$ on state-action tuples $(s,a)$ from the learned Q-function’s prediction. Formally, this means that we utilize $\hat{Q}(s, a) := Q(s,a) - \alpha D_\text{KL}(\pi(a|s),\pi^\beta(a|s))$ as our conservative Q-value estimate for BRAC.
>
> We evaluated BRAC + CDS on the Meta-World tasks and compared it to BRAC + Sharing All, since Sharing All was the second-best data sharing scheme on these tasks when CQL is used as the base offline RL algorithm. We use $\pm$ to denote the 95%-confidence interval. As observed below, BRAC + CDS significantly outperforms BRAC with Sharing All. This indicates that CDS is effective on top of both BRAC and CQL, two very different algorithms. We will evaluate CDS on top of BRAC on all the other domains, compare it to the other methods (e.g., No Sharing) and add the final results to the final version of the paper, including a discussion of how CDS can be used with other base offline RL methods.
>
> |Task|BRAC + Sharing All           |BRAC + CDS (Ours)|
> |----|-----------------------------|-----------------|
> |Meta-World (average task)| 40.2% $\pm$ 6.5%             | 50.8% $\pm$ 5.9% |
>
> Regarding implementation details of BRAC+CDS, the hyperparameter $\alpha$ that appears in the expression for the conservative Q-value estimate for BRAC discussed above, is identical to the hyperparameter $\alpha$ in Equation 7 of Wu et al. 2019, that controls the strength of behavior regularization. For both the Sharing All baseline and CDS, we utilize identical $\alpha=0.1$.
>
> **Clarity concerns:** We have now edited the paper to resolve the clarity concerns: in Equation 4, the $\pi_\beta^\mathrm{eff}$ on the LHS is used to denote $\arg\max$ from the second maximization over $\pi_\beta$ appearing in the RHS. We now clarify that the numbers in Table 2 denote the average return for the walker2d task and success rate for Meta-World and ant mazes. We will add an image of the ant maze in Figure 2 in the final.

---

> > ### Comment · Reviewer_BHsT · 2021-08-18
> > **Thanks for the author response**
> >
> > Thanks for additional experiments to show the general application of CDS. The results look very nice!

---

### Author Response · Authors · 2021-08-10
**Summary of changes**

We thank the reviewers for their feedback and a generally positive assessment of the paper. In this summary note, we would like to highlight our responses to the main concerns raised by the reviewers and the main experiments that we have added in the rebuttal period.

1. **[Reviewer ceUo, dWyG]** We have now run more seeds for each algorithm, totaling to 8 seeds for each method in the multi-task walker and Meta-World domains and 4 seeds for each method in the robotic manipulation domain from raw visual inputs. CDS still consistently outperforms other methods in all of the domains. Exact performance numbers can be found in the responses to **Reviewers ceUo** and **dWyG**.
2. **[Reviewer BHsT]** We implemented CDS on top of BRAC (a policy constraint method) and compared it to BRAC + Sharing All data sharing scheme, which is the second-best method on the Meta-World domain when using CQL. We find that BRAC + CDS outperforms BRAC + Sharing All, indicating that CDS can work over different kinds of offline RL methods. Exact performance numbers can be found in the response to **Reviewer BHsT**.
3. **[Reviewer ceUo]** To better understand how CDS shares data across tasks, we now provide three diagnostic visualizations/metrics on Meta-World, Antmaze and walker2d domains. In all cases, we find that CDS is able to filter out irrelevant/bad transitions for solving a target task.
4. **[Reviewer ceUo]** Per the reviewer’s suggestion, we compared CDS to representation learning (Yang et al. 2021) + offline multi-task RL on the learned representations. As shown in the performance table in our response to **Reviewer ceUo**, we find that data sharing via CDS outperforms the representation learning approach.

We would appreciate it if the reviewers can please take a look at these changes and let us know if they have any other concerns.

---

### Decision · Program_Chairs · 2021-09-27

**Decision:**

Accept (Poster)

**Comment:**

This paper presents a data-sharing strategy called "conservative data sharing" CDS to tackle the distributional shift of the offline RL algorithm in the multitask setting. All the reviewers evaluated positively about the work, and this would make a solid constribution to NeurIPS. I hope that reviewer comments help the authors for future work.